# Antihyperglycemic Effects of *Salvia polystachya* Cav. and Its Terpenoids: α-Glucosidase and SGLT1 Inhibitors

**DOI:** 10.3390/plants11050575

**Published:** 2022-02-22

**Authors:** Rocio Ortega, Miguel Valdés, Francisco J. Alarcón-Aguilar, Ángeles Fortis-Barrera, Elizabeth Barbosa, Claudia Velazquez, Fernando Calzada

**Affiliations:** 1Doctorado en Ciencias Biológicas y de la Salud, Universidad Autónoma Metropolitana-Iztapalapa, UAM-I, Mexico City CP 09340, Mexico; 2Av. San Rafael Atlixco 186, Leyes de Reforma 1ra Sección, Iztapalapa, Mexico City CP 09340, Mexico; 3Unidad de Investigación Médica en Farmacología, UMAE Hospital de Especialidades, 2° Piso CORSE Centro Médico Nacional Siglo XXI, IMSS, Av. Cuauhtémoc 330, Col. Doctores, Mexico City CP 06725, Mexico; 4Laboratorio de Farmacología, Departamento de Ciencias de la Salud, División de CBS, Universidad Autónoma Metropolitana-Iztapalapa, UAM-I, Av. San Rafael Atlixco 186, Leyes de Reforma 1ra Sección, Mexico City CP 09340, Mexico; aaaf@xanum.uam.mx (F.J.A.-A.); fortis11_10@yahoo.com.mx (Á.F.-B.); 5Escuela Superior de Medicina, Instituto Politécnico Nacional, Salvador Díaz Mirón esq. Plan de San Luis S/N, Miguel Hidalgo, Casco de Santo Tomas, Mexico City CP 11340, Mexico; rebc78@yahoo.com.mx; 6Área Académica de Farmacia, Instituto de Ciencias de la Salud, Universidad Autónoma del Estado de Hidalgo, Km 4.5, Carretera Pachuca-Tulancingo, Unidad Universitaria, Pachuca CP 42076, Mexico; cvg09@yahoo.com

**Keywords:** *Salvia polystachya*, antihyperglycemic activity, α-glucosidase inhibitor, SGLT1 inhibitor, acute oral toxicity, docking analysis, ursolic acid, oleanolic acid, diabetes mellitus

## Abstract

The antihyperglycemic activity of ethanolic extract from *Salvia polystachya* (EESpS) and its products was evaluated using in vivo, ex vivo and in silico assays; additionally, an acute toxicity assay was evaluated. EESpS was classified as a nontoxic class 5 drug. EESpS, ethyl acetate fraction (EtOAcFr), secondary-6-fraction (SeFr6), ursolic acid (UA), and oleanolic acid (OA) reduced the hyperglycemia in DM2 mice. α-glucosidase inhibition was evaluated with oral sucrose and starch tolerance tests (OSuTT and OStTT), an intestinal sucrose hydrolysis (ISH) assay and molecular docking studies using acarbose as control. SGLT1 inhibition was evaluated with oral glucose and galactose tolerance tests (OGTT and OGaTT), an intestinal glucose absorption (IGA) assay and molecular docking studies using canagliflozin as the control. During the carbohydrate tolerance tests, all the treatments reduced the postprandial peak, similar to the control drugs. During the ISH, IC_50_ values of 739.9 and 726.3 µM for UA and OA, respectively, were calculated. During the IGA, IC_50_ values of 966.6 and 849.3 for UA, OA respectively, were calculated. Finally, during the molecular docking studies, UA and OA showed ∆G values of −6.41 and −5.48 kcal/mol^−1^, respectively, on α-glucosidase enzymes. During SGLT1, UA and OA showed ∆G values of −10.55 and −9.65, respectively.

## 1. Introduction

Diabetes mellitus (DM) is a serious chronic disease that occurs when there is no optimal use of insulin in the organism due to a lack of insulin production or because the organism cannot properly use the insulin that is produced due to insulin resistance [1,2]. This alteration in insulin production and utilization is reflected in an increase in the concentration of blood glucose, known as hyperglycemia [3]. There are several drug families that are used as a treatment for DM, and all drug families are classified according to their mechanism pathway. Drugs classified as secretagogues include sulfonylureas and meglitinides [4], insulin sensitizers include biguanides and thiazolidinediones [5], and other drugs include glucagon-like peptide 1 (GLP-1) analogues [6], dipeptidyl peptidase-4 (DPP4) inhibitors [7], α-glucosidase [8,9], and SGLT inhibitors [10]. However, despite the therapeutic effect that is wanted after the administration of the treatments previously described, all of them have side effects such as hypoglycemia, gastrointestinal disorders, diarrhea, nausea, vomiting, abdominal discomfort, flatulence production, and urinary tract infection, among others [11]. One of the principal approaches for reducing postprandial hyperglycemia in the type 2 diabetes (T2D) population is the prevention of hydrolysis and the absorption of carbohydrates after food uptake. Therefore, the effective reduction of blood glucose levels after food uptake can be a key step in preventing or reversing diabetic complications and improving the life quality of T2D patients [12]. This reduction of hydrolysis and the absorption of carbohydrates can be reached with the inhibition of α-glucosidase and sodium–glucose cotransporter type 1 (SGLT1) enzymes [13]. In the first case, the α-glucosidase enzymes are located in the small bowel brush, and they degrade oligosaccharides such as sucrose and lactose to monosaccharides such as α-glucose by hydrolyzing the glycosidic bonds [14], delaying the absorption of carbohydrates since complex carbohydrates cannot be absorbed; thus, postprandial hyperglycemia can be delayed [14]. SGLT1 is highly expressed on the brush border membrane of enterocytes in the proximal part of the small intestine. It is responsible for glucose absorption. SGLT1 enzymes couple with the entrance of Na+ and glucose, facilitating the transport of glucose from the small intestine into the enterocyte; subsequently, the accumulated glucose in the enterocytes is mobilized out of them by glucose transport facilitating systems (GLUTs), and absorbed into the bloodstream [15]. Accordingly, both mechanisms of action are important points to take into account in the search for new molecules that help to reduce hyperglycemia in T2D patients.

One alternative which a major part of the population adopts is the use of medicinal plants; some of them are used for the treatment of several disorders and represent a potential source of valuable drugs in several pathologies, including T2D [16], a disease with the highest rates of prevalence and mortality worldwide [17,18]. Globally, the population uses medicinal plants to treat diabetes symptoms in an empirical way [19]. In this sense, several species of the Lamiaceae family have been proposed as α-glucosidase inhibitors, including *S. amarissima*, *S. urmiensis*, *S. miltiorrhiza*, *S. splendens*, *S. hypoleuca*, *S. fruticosa*, *S. syriaca*, *S. santolinifolia*, *S. moorcraftiana*, *S. limbata*, *S. atropatana*, *S. nemorosa*, and *S. multicaulis* [20,21,22,23,24]. Moreover, the diterpenes amarisolide, teotihuacanin, and amarissinins A and C are reported as the active principles in *S. amarissima* [25,26]. Moreover, in *S. reptans*, *S. microphylla* and *S. amarissima*, the presence of ursolic acid (UA), oleanolic acid (OA) and β-sitosterol [27,28,29] has been reported, which have shown significant inhibitory activity towards α-glucosidase [29].

The interest in the study of terpenes has increased because several authors have reported this kind of compound to have multiple biological attributes that can be useful for the treatment of several diseases; these biological attributes include antifungal, antibacterial, antiviral, antitumor, antiparasitic, anti-inflammatory, analgesic and hypoglycemic properties [30]. Considering the above, *Salvia polystachya* Cav. (Lamiaceae) is a herbaceous and perennial plant endemic to Mexico, known as “chía” [31,32]. It has been reported to have several properties, such as purgative, antipyretic, antimalarial, antihemorrhagic, and emollient effects, as well as having been reported as an effective treatment for heartburn and dysentery [33,34]. From the leaves, neo-clerodane diterpenes, such as polystachynes A–H, 15-epi-polystachine G, salvifiline A and C, 15-epi-salvifiline A, and linearolactone, and the flavone 3’,5,6,7-tetrahydroxy-4’-methoxyflavone, were isolated [33,34]. Among these, the diterpene linearolactone exhibited antiprotozoal activity in *Entamoeba histolytica* and *Giardia lamblia* [35]; moreover, polystachynes B and G, linearolactone and 15-epi-salvifiline A increased the gene expression of elastin and the type I, III, and V of collagens [36]. Although several species of the Salvia genus have been reported to have antidiabetic properties, there are no scientific reports that support *S. polystachya*’s effects on hyperglycemia yet. Thus, considering the chemotaxonomical criteria, the aim of our study was to explore the potential antidiabetic effect of *S. polystachya* and its isolated products, as well to evaluate its effect as an α-glucosidase and SGLT1 inhibitor using activity-guided fractionation as a strategy, using in vivo, ex vivo and in silico assays.

## 2. Results

### 2.1. In Vivo Assays

#### 2.1.1. Acute Oral Toxicity of the Ethanolic Extract of the Stems from *S. polystachya*

Acute oral toxicity assays of the extract from *S. polystachya* stems were performed according to Guideline 423 of the OECD [37]. It was found that 4 h after the administration of the treatments, the extract did not generate alterations in the animals. Moreover, EESpS did not generate mortality at the doses evaluated. At the end of the test (14 days), the surviving mice did not display body weight loss compared with normal mice (Table 1).

#### 2.1.2. Acute Effect of Ethanolic Extract of Stems from *S. polystachya* and its Products over Glycemia in Normal and Diabetic Mice

Considering that EESpS did not show toxicity, the acute evaluation in diabetic mice was carried out. The doses selected were 50, 100, and 200 mg/kg^−1^. The results showed that the dose with the best effect on hyperglycemia was 100 mg/kg^−1^. This effect was similar to that observed in the groups treated with the control drugs acarbose and glibenclamide (Table 2). Acarbose, an α-glucosidase inhibitor, was used as the comparative control. In the case of glibenclamide, it was used to demonstrate pancreatic β-cell viability in streptozocin-induced diabetic mice in agreement with the T2D model.

Once its activity over hyperglycemia was demonstrated, EESpS was subjected to partition, obtaining two fractions, AqRFr and EtOAcFr, which were also evaluated in diabetic mice (Table 2). Both fractions significantly reduced blood glucose from 30 to 120 min; this activity was similar to that of the control drug, acarbose. Nevertheless, EtOAcFr was selected to continue the study because its effect on hyperglycemia was slightly greater compared with the observed effect in AqRFr.

EtOAcFr was subjected to secondary fractionation using column chromatography; ten secondary fractions (SeFr1–SeFr10) were obtained, and they were evaluated in diabetic mice. The secondary fraction SeFr6 was the only one with a significant effect on the hyperglycemic values at 120 min (Table 2). SeFr6 was compared with 42 standard compounds of different polarities including polystachynes A, B, and D, as well as linearolactone, which was previously isolated from *S. polystachya* [35]. The analysis of SeFr6 was performed using high-performance liquid chromatography with diode array detection (HPLC-DAD), and the standards of the ursolic acid and oleanolic acid were used. Figure 1 shows the HPLC chromatograms of EtOAcFr, SeFr6, UA and OA. The complete analysis of the chromatogram of SeFr6 (Figure 1a) showed the presence of UA at 8.008 min (17.26%) and OA at 8.366 min (47.28%); these were compared with the chromatograms of UA and OA standards (Figure 1c and 1d, respectively).

The compounds identified, UA and OA, were evaluated in diabetic mice (Table 2). OA significantly decreased the blood glucose levels after 90 min, until 120 min. In the case of UA, this compound significantly decreased blood glucose levels after 30 min, until 120 min; this effect was similar to glibenclamide.

#### 2.1.3. Oral Sucrose and Starch Tolerance Tests of the Products Obtained from Stems from *S. polystachya*

During the OSuTT assay, all the treatments reduced the postprandial glucose peak. It is important to mention that EESpS, EtOAcFr (Figure 2A), OA, and UA (Figure 2B) significantly reduced the hyperglycemia peak observed 30 min after the sucrose load, with percentages of the inhibition of the peak of 30.6, 47.8, 29.8 and 30.7%, respectively. The activity demonstrated by the products obtained from the stems of *S. polystachya* was less than the activity observed after the administration of the control drug acarbose, an α-glucosidase inhibitor, which significantly reduced the postprandial peak with 100% inhibition at 30 min.

In OStTT, a similar effect to OSuTT was observed. EESpS significantly reduced the postprandial peak during the assay with a 32.9% reduction at 30 min. In the case of EtOAcFr, there was only a significant reduction at 30 min with a 13.6% reduction (Figure 3A), in comparison with the starch group. When UA and OA were administered, a significant reduction in the postprandial peak was observed during all the assays with a 32.6 and 36% reduction at 30 min, respectively. The control drug, acarbose, also reduced the postprandial peak, with a 25.9% reduction at 30 min. It is important to mention that UA and OA showed a greater inhibitory effect than the control drug at 30 min (Figure 3B).

#### 2.1.4. Oral Glucose and Galactose Tolerance Tests of the Products Obtained from Stems from *S. polystachya*

EESpS, EtOAcFr, UA and OA were evaluated in oral glucose (OGTT) and galactose (OGaTT) tolerance tests, using canagliflozin, an inhibitor of SGLT1/2 cotransporters, as a control drug. EESpS and EtOAcFr significantly reduced the glucose postprandial peak at 30 min. Their effect was similar to what was observed in the control drug canagliflozin (Figure 4A). When UA and OA were tested in OGTT, a significant reduction in the postprandial peak at 30 min in comparison with the glucose group was observed. However, this effect was lower than what was observed after the administration of canagliflozin (Figure 4B).

When the OGaTT was performed, the group treated with this carbohydrate showed a postprandial peak from 30 min and until the end of assay. The groups treated with EESpS and EtOAcFr showed a significant reduction in the postprandial peak from 60 min until the end of the assay (Figure 5A). Additionally, UA and OA significantly reduced the postprandial peak from 60 min until the end of the assay (Figure 5B). In all the treatments, the effects observed were less than the group treated with canagliflozin.

Once evaluated, the products in the in vivo assays were evaluated in ex vivo assays in order to determine if the antidiabetic activity was related to a reduction in intestinal sucrose hydrolysis as well as intestinal glucose absorption.

### 2.2. Ex Vivo Assays

#### 2.2.1. Inhibition of Intestinal Sucrose Hydrolysis of the Products Obtained from the Stems from *S. polystachya*

After the addition of the treatments in the intestine portions, the EESpS, EtOAcFr and UA showed a significant reduction in the glucose absorbed at concentrations of 200, 400 and 800 µgmL^−1^ with half-maximal inhibitory concentration (IC_50_) calculated at 734.3, 573.5 µgmL^−1^, and 739.9 µM, respectively. When OA was evaluated, it showed a significant reduction in glucose absorbed at 400 and 800 µM with an IC_50_ of 726.3 µM. Finally, acarbose showed a significant reduction in glucose absorbed at 200, 400 and 800 µM, and the IC_50_ calculated was 173 µM (Table 3).

#### 2.2.2. Inhibition of Intestinal Glucose Absorption of the Products Obtained from the Stems from *S. polystachya*

When the treatments were added to the intestine portions in IGA, the EEsPS significantly reduced the quantity of glucose absorbed at 400 and 800 µgmL^−1^ with a calculated IC_50_ of 1536.3 µgmL^−1^. In the case of EtOAcFr, it showed a significant reduction in the glucose absorbed at concentrations of 200, 400 and 800 µgmL^−1^ with a calculated IC_50_ of 697.3 µgmL^−1^. In the case of ursolic acid, it showed a significant reduction in glucose absorbed at 400 and 800 µM with an IC_50_ of 966.6 µM. The oleanolic acid showed a significant reduction in glucose absorbed at 200, 400 and 800 µM, and the IC_50_ values calculated were 849.3 and 834.5 µM, respectively. Finally, canagliflozin showed a significative reduction in glucose absorbed at 400 and 800 µM, with a calculated IC_50_ of 834.5 µM (Table 4).

The results obtained are according to the in vivo OSuTT, OStTT, OGTT and OGaTT evaluations of oleanolic acid and ursolic acid that suggest the inhibition of the a-glucosidase enzymes and also a reduction in the glucose absorption mediated by the inhibition of SGLT-1. Additionally, in silico studies were carried out in order to give additional support to the previous in vivo and ex vivo results.

### 2.3. In Silico Assays

#### 2.3.1. Molecular Docking Studies of Ursolic Acid (UA), Oleanolic Acid (OA) and Acarbose on α-Glucosidase Enzyme

In order to show the possible interaction of UA and OA, a molecular docking study was carried out using as a target the α-glucosidase enzyme, which is involved in the control of complex carbohydrate hydrolysis (the crystal structure of human lysosomal acid-α-glucosidase was used: RCSB, PDB ID: 5NN8). Moreover, acarbose was used as a control, and its binding site was compared with the UA and OA binding sites.

According to the results obtained, OA showed a greater affinity than UA to the binding site, with ∆G values of −6.41 kcal-mol^−1^ and −5.48 kcal-mol^−1^, respectively. OA and UA showed they had one polar interaction with each ligand, OA with Leu677, and UA with Asp518 residues (Table 5). In the case of acarbose, it showed the best affinity to the α-glucosidase enzyme, with a ∆G value of −8.33 kcal-mol^−1^. Moreover, this ligand showed eight polar interactions with Arg281, Asp282, Ala284, Arg600, Asp616, Gly651, Ser676 and Leu678. It is important to mention that the three ligands showed the same binding pocket; in the case of OA and acarbose, it showed a similar binding position. In the case of UA, this ligand binds in a different position than OA and acarbose (Figure 6).

#### 2.3.2. Molecular Docking Studies of Ursolic Acid (UA), Oleanolic Acid (OA) and Canagliflozin on SGLT1 Cotransporter

The second molecular docking study was carried out using the SGLT1 cotransporter as a target. SGLT1 is involved in the control of glucose absorption (the crystal structure of human sodium/glucose cotransporter, UniProt ID: P13866). Canagliflozin was used as a control in order to compare the binding site with UA and OA.

UA showed an affinity in a different site than canagliflozin (Figure 7), with a ∆G of −9.65 kcal-mol^−1^. The interaction of UA with this binding site showed two polar interactions with Gln451 and Gly523. In the case of OA, this ligand showed the same binding position as UA. Additionally, OA had a ∆G of −10.55 kcal-mol^−1^ with only one polar interaction with Gln 451. On the other hand, canagliflozin showed the best affinity to SGLT1 with a ∆G of −11.04 kcal-mol^−1^; this ligand showed five polar interactions with Met283, Thr287, Tyr290, Trp291 and Gln 457 (Table 5).

## 3. Discussion

Several species from the Lamiaceae family have been proposed to have an antidiabetic effect, ursolic acid, oleanolic acid and β-sitosterol being the products related with this activity [25,26]. In the case of ursolic acid and oleanolic acid, they have been described as potential α-glucosidase inhibitors using in vitro and in silico experiments [29]. Considering the above and taking into account the constant need for the development of new drugs to treat the hyperglycemia caused by diabetes mellitus, the potential antidiabetic effect of *S. polystachya* (a species that belongs to the Lamiaceae family) was evaluated, in consideration of a chemotaxonomic criterion. The evaluation of the ethanolic extract obtained from the stems of *S. polystachya* was carried out using activity-guided fractionation as a strategy, with in vivo, ex vivo and in silico assays.

Initially, the toxicological study (according to the OCDE’s Guideline 423) of the extract obtained was carried out, and the results showed that the ethanolic extract of the stem from *S. polystachya* did not cause behavior alterations, visible tissue damage, modifications in the body or organ weight. According to the Globally Harmonized Classification System for chemical substances and mixtures (GSH) adopted by the OECD [37], the extract is classified as a nontoxic class 5 drug, and is considered a nontoxic substance [37].

Once the toxicity of our sample was evaluated, the evaluation in diabetic mice was carried out. The experimental diabetes mellitus type 2 (DM2) was induced using the streptozocin–nicotinamide (STZ–NA) model [13]. The administration of STZ–NA induces diabetes due to selective pancreatic β-cell cytotoxicity by STZ [38,39], followed by β-cell protection generated by the administration of NA [40]. Finally, the model results in a partial inhibition of insulin secretion [41]. Therefore, the NA–STZ-induced diabetes model is characterized by hyperglycemia and glucose intolerance, such as in T2D. It is a valuable model for the study of potential antidiabetic drugs, including medicinal plants [23,27,42]. The administration of the extract from the stems from *S. polystachya* (EESpS) generated a significative reduction in hyperglycemia in the animals with experimental DM2. These results are similar to those reported in other vegetal species from the Lamiaceae family [25,26,27,28,29,30], where the authors describe the diminution of hyperglycemia after a single administration of ethanolic extracts from the leaves of other Lamiaceae species. Moreover, in some studies, a subchronic administration of extracts results in an adequate control of hyperglycemia accompanied by a reduction in the glycated hemoglobin percentage [20].

EESpS was fractionated and the obtained fractions were also evaluated in diabetic mice, with the result that EtOAcFr significantly reduced the hyperglycemia in the DM2 model. Thus, this fraction was submitted to a secondary fractionation resulting in ten secondary fractions, of which SeFr6 was the most active. To identify the compounds responsible for the observed effects of the *S. polystachya* stem extract, a phytochemical screening from EtOAcFr was performed. HPLC–diode array detection (DAD) allowed us to determine the presence of ursolic and oleanolic acid, two terpenoids that have been described as α-glucosidase inhibitors in in vitro and in silico diabetic models [29]. In this sense, 42 terpenoids previously reported in Salvia species were checked in EtOAcFr and SeFr6, including polystachynes A, B, and D, as well as linearolactone, which was previously isolated from *S. polystachya* [35]; however, only the presence of ursolic acid and oleanolic acid (UA and OA, respectively) was identified. It is important to mention that the isolation of UA and OA in this vegetal species has not been reported yet. When these products were evaluated in the DM2 model, they showed significant reductions in blood glucose levels. Some authors have reported that UA acted as an effective insulin mimetic and as an insulin sensitizer [43], increasing insulin vesicle translocation, insulin secretion and augmented glycogen content [44]. Based on these reports, the effect observed in EESpS could be partially explained. To continue the study, the reduction of complex carbohydrate hydrolysis and the absorption of simple carbohydrates mediated by α-glucosidases and SGLT1, respectively were addressed.

The α-glucosidase inhibitors can delay the liberation of glucose from dietary complex carbohydrates, retarding glucose absorption and lowering the postprandial blood glucose peak [45]. In the present research, the α-glucosidase inhibitory effect was studied using oral sucrose and starch tolerance tests (OSuTT and OstTT, respectively). In these assays, the postprandial glucose peak after the complex carbohydrate load was measured to determine the possible inhibition of these enzymes which are involved in the hydrolysis of different types of glycosidic bonds (sucrose: α-1,2; starch: α-1,4) then EESpS EtOAcFr. UA and OA significantly reduced the postprandial peak after sucrose or starch administration, possibly due to an inhibitory effect on α-glucosidase, preventing hyperglycemia. However, to corroborate the results obtained, ex vivo and in silico studies were carried out.

The second mechanism of action evaluated was that of the type 1 inhibitor of sodium–glucose cotransporters (SGLT). These cotransporters are part of a subgroup of the solute carrier group (SLC5), which includes six members that differ in their preferences for sugar binding, and all the members of this family use the electromechanical gradient of sodium to transport sugar molecules against a chemical gradient into cells. Among the most studied of the SLC5 family, SGLT1 and 2 are highlighted. SGLT2 is expressed mostly in early renal proximal tubules and is responsible for >90% of the renal reabsorption of filtered glucose (160–180 g/day) and SGLT1 plays an important role in glucose absorption from the intestinal lumen into the epithelial cells of the small intestine [46]. In order to determine the effect of the products isolated from the stems of *S. polystachya*, oral glucose and galactose tolerance tests were carried out (OGTT and OGaTT, respectively) with the objective of determining whether the treatments inhibited the SGLT1 expressed in the membrane of enterocytes, which are responsible for mediating intestinal glucose absorption [47,48,49]. It is possible that the reduction of the postprandial peak after the carbohydrate load observed in EESpS, EtOAcFr, UA and OA treatments works in conjunction with the inhibition of α-glucosidases shown in the previous assays. Moreover, in the case of EESpS and EtOAcFr, one or more compounds might inhibit some of the glucose transporters involved in its absorption, with UA and OA being partly responsible for the effects observed. Additional studies to identify other compounds and determine their possible synergic action are mandatory. These results support that *S. polystachya* contains compounds with the capacity to inhibit α-glucosidases.

The next step was conducting the ex vivo assays. These studies were carried out with the objective of corroborating the activity demonstrated in the in vivo assays. First, in the intestinal sucrose hydrolysis assay (ISH), all the products significantly reduced the quantity of glucose in the aqueous external medium. This can be interpreted as a possible inhibition of the α-glucosidases present in the portion of intestine used. The IC_50_ calculated in all the treatments showed the potential inhibitory effect of the hydrolysis of complex carbohydrates mediated by α-glucosidases. Despite acarbose showing a minor IC_50_ value, the results shown by the products isolated from the stems of *S. polystachya* are important. Authors describe that UA and OA isolated from *Salvia africana-lutea* exhibited promising α-glucosidase inhibitory activity, and the inhibition kinetic analysis showed IC_50_ values in the range of 24.7–188.7 µM compared to 945.5 µM for acarbose. It has been proposed that the presence of a methyl group at the C-19 position had a positive effect on the inhibitory activity of UA [50].

The absorption of simple carbohydrates was evaluated with the intestinal glucose absorption assay (IGA). After the addition, all the products showed a significant reduction in the total glucose absorbed. This result corroborates the activity observed in OGTT and OGaTT. In regard to the IC_50_ obtained for all the treatments, UA and OA (966.6 and 849.3 µM, respectively) were similar to the IC_50_ obtained with the control drug canagliflozin (834.5 µM). The in vivo and ex vivo results suggest that the antihyperglycemic activity observed after the administration of the products obtained from the stems of *S. polystachya* can be mediated by the inhibition of the hydrolysis of complex carbohydrates and the absorption of simple carbohydrates.

Finally, in silico studies were carried out with the objective of determining the possible binding site of UA and OA in the α-glucosidases and SGLT1 cotransporter. In the case of the enzyme α-glucosidase, UA showed a ∆G value of −5.48 kcal-mol^−1^; moreover, this product shares some binding amino acid residues (Asp616 and Leu650) with acarbose. Despite the binding position not being the same (Figure 6), this position may help to avoid α-glucosidase activity. Perhaps, as some authors describe, the methyl group at the C-19 position could be involved in the binding position [50]. Molecular dynamic studies need to be carried out to observe if this binding position avoids the interaction between sucrose and the α-glucosidase enzyme. In the case of OA, it showed a ∆G of −6.41 kcal-mol^−1^, and shares six binding amino acid residues (Asp282, Met519, Leu650, Gly651, Ser676 and Leu678) with acarbose. It is important to mention that the binding position is very similar to acarbose. This may explain the results obtained during the in vivo OSTT and OStTT and the ex vivo ISH assays, where OA reduced the postprandial peak of glucose, and also reduced the quantity of glucose absorbed in ISH with a lower CE_50_ (726.3 µM) than that obtained in UA (739.9 µM).

In regard to molecular docking studies with SGLT1, UA (∆G of −9.65 kcal-mol^−1^) and OA (∆G of −10.51 kcal-mol^−1^), each showed important binding values; however, the binding position was not similar to that obtained in the molecular docking of canagliflozin (∆G of −11.04 kcal-mol^−1^), which was shown to bind in a different place in the SGLT1 cotransporter. According to the results obtained during the in vivo and ex vivo studies, AU and OA reduced the postprandial peak of glucose in OGTT and OGaTT, and also showed an important CE_50_ (966.6 and 849.3 µM, respectively). There is the possibility that they inhibit the SGLT1 activity in an allosteric site. However, the next step of the investigation is to perform molecular dynamic studies accompanied by enzyme inhibition studies in order to confirm the results obtained.

UA and OA are pentacyclic triterpenoids, isomers that differ in the position of a methyl residue connected to the C-19 or C-20 in the E ring. These modifications result in different pentacyclic triterpenes; the former is oleanane, whereas the latter is ursane [51,52]. These compounds naturally occur in many species of plants; in particular, they have been reported in some plants of the Salvia genus. The OA from the *S. moorcraftiama* has been reported to have the potential to reduce hyperglycemia and the complications of diabetes [23,31,53,54], and have great permeability in the small bowel [55,56,57,58], which were proposed as α-glucosidase inhibitors [59,60,61,62]. The present study demonstrates for the first time the isolation of UA and OA in *S. polystachya*. In addition, this study evaluates their antihyperglycemic activity with a possible dual inhibitory activity effect over α-glucosidase enzymes and the SGLT1 cotransporter.

## 4. Materials and Methods

### 4.1. General Information

Ethanol anhydrous (CC:15568604) and ethyl acetate (CC:10382681) were purchased from J.T. Baker (Thermo Fisher Scientific, Waltham, MA, USA). Streptozotocin (≥75% α-anomer basis, PN: S0130-5G), nicotinamide (≥99.5%, PN: 47865-U), ursolic acid (≥90%, PN: U6753-500MG), oleanolic acid (≥97%, PN: O5504-500MG), sucrose (≥99.5% GC, PN: S9378-1Kg), acetonitrile HPLC gradient grade (≥ 99.9, PN: 34851-1L) and ethanol HPLC gradient grade (100%, PN: 459828-1L) were purchased from Sigma-Aldrich (St. Louis, MI, USA). Acarbose (Glucobay, tablets of 50 mg, Bayer Mexico S.A. DE C.V.), canagliflozin (Invokana, tablets of 300 mg, Janssen-Ortho LLC, Puerto Rico) and glibenclamide (Glibenclamide, tablets of 5 mg, Silanes, Mexico) were purchased from the local pharmacy. Buffer solution (citric acid/sodium hydroxide/hydrogen chloride, pH 4.00, CC: 109445). Saline solution 0.9% (solution 1000 mL) and DX-5 glucose solution 5% (solution 500 mL) were purchased from PISA Pharmaceutics (Pisa, Mexico City, Mexico).

### 4.2. Plant Material

*S. polystachya* stems were collected in San Gregorio Atlapulco, Xochimilco (19°14′9.919″ N 99°2′53.879″ W), Mexico City, Mexico. The plant was identified by the M.Sc. Santiago Xolalpa in the Herbarium of the Mexican Institute of Social Security (IMSSM-Herbarium), with voucher specimen number 16386. The sample was cleaned and air-dried at ambient temperature, and finally, the dried samples were ground using a laboratory grinder (model M-22-RW, Fundición Torrey, Apodaca, Nuevo León, México).

### 4.3. Preparation of Ethanolic Extracts, Fractionation, and Characterization of Ursolic and Oleanolic Acid

The stems (695.1 g) from *S. polystachya* were extracted with EtOH (6 L × 3) for one week, and filtered (Whatman No.1). The extract filtered was evaporated to dryness using a rotary evaporator at 35 °C (Buchi, Flawil, Switzerland), to obtain 16.9 g of ethanolic extract from stems (EESpS, 2.4% yield); this was subjected to biological assays. Once the antihyperglycemic activity of EESpS was examined, it was submitted for partitioning. Briefly, a portion of EESpS (50 g) was suspended in 10% EtOH–water (100 mL) and successively partitioned with EtOAc (150 mL × 2) to obtain 19.8 g of EtOAc fraction (EtOAcFr). The aqueous residual layer was collected to obtain 28.9 g of aqueous residual fraction (AqRFr).

The antihyperglycemic activity was associated with EtOAcFr, then a portion (450 mg) was submitted for separation by open column chromatography (2 cm × 34.5 cm), packed with silica gel 60 (70–230 mesh), and eluted with solvents of crescent polarity: hexane, hexane/ethyl acetate, and ethyl acetate/methanol. Ten secondary fractions were obtained: SeFr1 (26. 9 mg), SeFr2 (13.1 mg), SeFr3 (218 mg), SeFr4 (57.4 mg), SeFr5 (128.4 mg), SeFr6 (194.9 mg), SeFr7 (49.4 mg), SeFr8 (12.3 mg), SeFr9 (123.4 mg), and SeFr10 (111.4 mg).

SeFr6 was the fraction with the best antihyperglycemic activity. It was then analyzed using HPLC-diode array detection (DAD) (Waters Agilent, 5301 Stevens Creek Blvd Santa, Clara, CA 95051, USA). The analysis was performed using an HPLC-DAD Waters 2795 liquid chromatograph system coupled with a Waters 996 photodiode array detector and an analytical Millennium 3.1 workstation equipped with a C18 analytical column (Waters, Mexico City, Mexico) with dimensions of 250 mm × 4.6 mm and a particle size of 5 μm (Spherisorb S50D52, Waters Corporation, Milford, MA, USA). For the analysis, 50 mg of the SeFr6 was dissolved in 10 mL of EtOH, and 20 μL of the sample was injected. For elution, a system comprising a binary mobile phase of acetonitrile 100% (solvent A) and acetic acid 2% (solvent B) in water was used. The chromatograph’s operating conditions were programmed to give the following linear gradient of 80 (A)/20 (B) for 15 min with a flow rate of 0.8 mLmin^−1^ of the mobile phase. The detections were made at a wavelength (λ) from 200 to 400 nm at room temperature and a total elution time of 25 min. At the end, the data collected were plotted, and the chromatograms shown in the results section show an absorbance of 220 nm, due to the samples only showing an absorbance at this wavelength. The presence of substances in the SeFr6 fraction was confirmed by comparing the retention times with the standards library. The reference standards of the ursolic acid (UA) and oleanolic acid (OA) used had a purity degree of 90 and 97, respectively. These were prepared and analyzed separately under the same conditions described above. In all cases, the water used was of HPLC quality and purified in a Milli-Q system (Millipore, Bedford, MA, USA).

### 4.4. Experimental Animals

Male and female BALB/c mice (21 ± 3 g) provided by the animal center at the 21st Century National Medical Center laboratory at the Mexican Institute of Social Security (CMN-SXXI-IMSS) were used. The animals were maintained at 22 ± 1 °C, with light/dark cycles of 12 h, and free access to water and food (standard rodent diet, LabDiet Formulab Diet 5008). The studies in rodents were performed in conformity with the Mexican Official Rule for Animal Care and Handling NOM-062-ZOO-1999 [63]. All investigations were conducted with the approval of the Specialty Hospital Ethics Committee of Centro Médico Nacional Siglo XX at IMSS (register: R-2020-3601-007).

### 4.5. In Vivo Assays

#### 4.5.1. Acute Toxicity Study

The acute oral toxicity study of the extract from the stems from S. *polystachya* was conducted in compliance with OECD’s Guideline 423 (Organization for Economic Cooperation and Development) [37]. BALB/c female mice (20–30 g), with free access to water and that were fasted overnight, were used. Animals were grouped as follows: the control group, which was treated with the vehicle (2% Tween 80 in water), and three groups treated with EESpS at 50, 300 and 200 mg/kg^−1^. The extract was dissolved in 2% Tween 80 in water, and all treatments were administered per os with an esophageal cannula. After the administration of the treatments, the animals were observed for 4 h and then for 14 days in order to record possible toxic effects, such as changes in behavior, body weight, urination, food intake, water intake, respiration, temperature, and eye and skin color, as well as convulsions, tremors, constipation, among others. At the end of the study (day 14), the animals were sacrificed, and the internal organs (stomach, gut, kidney, liver, and pancreas) were extracted, weighed, observed macroscopically and compared against the control group. The organs’ relative weights (ORW) were measured in accordance with the ORW formula [64].
$$Organ\;Relative\;Weight=\frac{Absolute\;weight\;\left(g\right)}{Mouse\;body\;weight\;at\;day\;14}\times 100$$

#### 4.5.2. Induction of Experimental Type 2 Diabetes

Experimental diabetes was induced by the streptozocin–nicotinamide (STZ–NA) model [12,13,20]. On the first day, mice fasted overnight (16 h) were used, and they intraperitoneally received (IP) 100 mg/kg^−1^ of STZ dissolved in a cold pH 4 buffer solution, and 30 min later, 240 mg/kg^−1^ of NA dissolved in saline solution were administered IP. Once STZ and NA were administered, the animal food was put out. On day 3, fasted mice (16 h) were administered a second dose of 100 mgkg^−1^ STZ. After administration, the animal food was put out and a 10% sucrose solution was used over two days. On day 5, the sucrose solution was withdrawn and substituted with water ad libitum. Then, 72 h later, the glycemia was measured by an enzymatic glucose oxidase method, using a glucometer (Evolution, Infopia USA, LLC, Titusville, FL, USA) [20]. Those animals with blood glucose levels between 250–350 mg/dL were considered for experiments.

#### 4.5.3. Acute Effect of Ethanolic Extract from Stems of *S. polystachya* and Its Products over Glycemia

This assay was conducted in diabetic mice (DM), and the animals were randomly divided into 12 groups (*n* = 6), as follows: normal mice (NM) and DM control groups, both treated with the vehicle, (2% Tween 80 in water); and ten groups treated with EESpS (50, 100 and 200 mg/kg^−1^), AqRFr, EtOAcFr, SeFr6, ursolic acid (UA) and oleanolic acid (OA) (50 mg/kg^−1^). In order to compare the antihyperglycemic activities of the products isolated from *S. polystachya*, the control drugs glibenclamide and acarbose (50 mg/kg^−1^) were used. All treatments were dissolved in Tween 80 (2% in water) as the vehicle and given in a single oral administration. Blood samples were obtained by puncturing the vein caudal before, and 0, 30, 60, 90 and 120 min after the administration of the treatments. Blood glucose levels (mg/dL) were determined by the enzymatic glucose oxidase method (Evolution, Infopia USA, LLC) [20].

#### 4.5.4. Oral Sucrose and Starch Tolerance Tests of the Products Obtained from Stems from *S. polystachya*

Oral sucrose and starch tolerance tests (OSuTT and OStTT, respectively) were conducted in male normoglycemic fasted mice. These were randomly divided into seven groups (*n* = 6), as follows: the vehicle group treated with the vehicle (2% Tween 80 in water); the sucrose group treated with the vehicle + sucrose (3 g/kg^−1^); five groups treated with EESpS (300 mg/kg^−1^), EtOAcFr (200 mg/kg^−1^), OA and UA (50 mg/kg^−1^); and a group treated with acarbose (50 mg/kg^−1^), an α-glucosidase inhibitor, used as the pharmacological control. All samples were solubilized with 2% Tween 80 in water and administered orally. Thirteen minutes after the administration of treatments, a sucrose load (3 g/kg^−1^) was administered to the groups. Once administered, the animals were maintained in metabolic cages during the assay. Blood glucose levels were measured by puncturing the caudal vein before the administration of treatments (0 h), and 30, 60, 90, and 120 min after administration. Blood glucose levels (mg/dL) were determined by the enzymatic glucose oxidase method (Evolution, Infopia USA, LLC) [20].

#### 4.5.5. Oral Glucose and Galactose Tolerance Tests of the Products Obtained from the Stems from *S. polystachya*

Oral glucose and galactose tolerance tests (OGTT and OGaTT, respectively) were carried out in the same conditions as the OSuTT and OStTT. The grouping was the same, with the exception that in OGTT and OGaTT, canagliflozin (50 mg/kg^−1^) was used as the pharmacological control, and a glucose load (1.5 g/kg^−1^) was used. The blood collection and measurement were carried out following the same conditions as OSuTT and OStTT.

### 4.6. Ex Vivo Assays

#### Determination of the Inhibition of Intestinal Sucrose Hydrolysis and Glucose Absorption of the Products Obtained from the Stems from *S. polystachya* Cav. and Its Products

The intestinal sucrose hydrolysis (ISH) and intestinal glucose absorption (IGA) assays were conducted according to Valdes et al. [13]. Male Sprague–Dawley rats were used for the assays. The rats were sacrificed according to NOM0062-ZOO-1999. The proximal small intestine (SI) was removed and the first portions of the SI (jejunum and duodenum) were cut into 3 cm portions. These portions were tied on their ends with a nonabsorbable silk suture (Ethicon, Johnson & Johnson, Somerville, MA, USA). For the ISH assay, the group treatments (*n* = 6) were prepared as follows: EESpS, EtOAcFr (200, 400 and 800 µg/mL^−1^), ursolic acid, oleanolic acid, and acarbose (200, 400 and 800 µM). The samples were dissolved in 1.5 mL of 15% sucrose solution as a vehicle. Additionally, a control group was treated only with the vehicle. All the treatments were injected with an insulin syringe inside the 3 cm SI portions previously made in a 0.5 mL volume. Immediately, SI portions were placed in a Petri dish with 15 mL of distilled water as the external aqueous medium (EAM) and incubated for 2 h at 37 °C with constant agitation. The quantities of glucose absorbed in the SI were measured in the EAM 1 h after adding the treatments using the glucose oxidase method.

In the case of the IGA assay, it was conducted under similar conditions to the ISH test, with the exception that canagliflozin (200, 400 and 800 µM) was used as the pharmacological control, the samples were dissolved in 1.5 mL of 5% glucose solution, and the measurements were carried out 1h after the incubation of the intestines. Finally, during the ISH and IGA assays, the results were compared and normalized with the control group at the different measurement times and the half maximal inhibitory concentrations (IC_50_) were calculated.

### 4.7. In Silico Assays

The chemical structure of the ligands ursolic acid (CID: 64945), oleanolic acid (CID: 10494) canagliflozin (CID: 24812758) and acarbose (CID: 41774) were retrieved from the chemical library PubChem (https://pubchem.ncbi.nlm.nih.gov/) (accessed on 6 November 2021); these were optimized and submitted to energetic and geometrical minimization using the Avogadro software [65]. Two different targets involved in the control of hyperglycemia were used, α-glucosidase (crystal structure of human lysosomal acid-α-glucosidase was used (RCSB, PDB ID: 5NN8)) and the SGLT-1 (crystal structure of human sodium/glucose cotransporter, UniProt ID: P13866) enzyme. These were retrieved from the Protein Data Bank (http://www.rcsb.org/ (accessed on 29 November 2021)) and UniProt database (https://www.uniprot.org) (accessed on 6 November 2021). The total molecules of water and ions that were not needed for catalytic activity were stripped to preserve the entire protein. All polar hydrogen atoms were added, ionized in a basic environment (pH = 7.4), and Gasteiger charges were assigned. The computed output topologies from the previous steps were used as input files for docking simulations.

The molecular docking experiments were carried out using AutoDock 4.2 software [66], and the search parameters were as follows: a grid-base procedure was employed to generate the affinity maps delimiting a grid box of 126 × 126 × 126 Å3 in each space coordinate, with a grid point spacing of 0.375 Å. The Lamarckian genetic algorithm was employed as a scoring function with a randomized initial population of 100 individuals and a maximum number of energy evaluations of 1 × 107 cycles. The analysis of the interactions in the enzyme/inhibitor complex was visualized with PyMOL software (the PyMOL Molecular Graphics System, Ver 2.0, Schrödinger, LLC). The validation of the molecular docking was carried out by re-docking the co-crystallized ligand in the receptors. The lowest energy pose of the co-crystallized ligands was superimposed and it was observed whether it maintained the same binding position. The RMSD were calculated and a reliable range within 2 Å is reported.

### 4.8. Statistical Analysis

Data are expressed as means ± standard error of the mean (SEM). The statistical analyses were determined using GraphPad Prism software (version 8.0.2, GraphPad Inc., La Jolla, CA, USA). One-way ANOVA evaluations were carried out followed by a post hoc Dunnett test. In all the cases, *p* < 0.05 was considered a statistically significant difference between the mean groups.

## 5. Conclusions

EESpS of *S. polystachya*, EtOAcFr, SeFr6 and the UA and OA identified from EtOAcFr reduced blood glucose levels in diabetic mice. This activity was confirmed through in vivo, ex vivo and in silico studies. The complete results of the analysis suggest that the antidiabetic activity of the products from the stems from *S. polystachya* is mediated in part by α-glucosidase enzyme inhibition and SGLT1 cotransporter inhibition. This research supports the phytochemical and pharmacological bases of *S. polystachya* and its use as a source of potential antidiabetic agents for T2D control.

## Figures and Tables

**Figure 1 plants-11-00575-f001:**
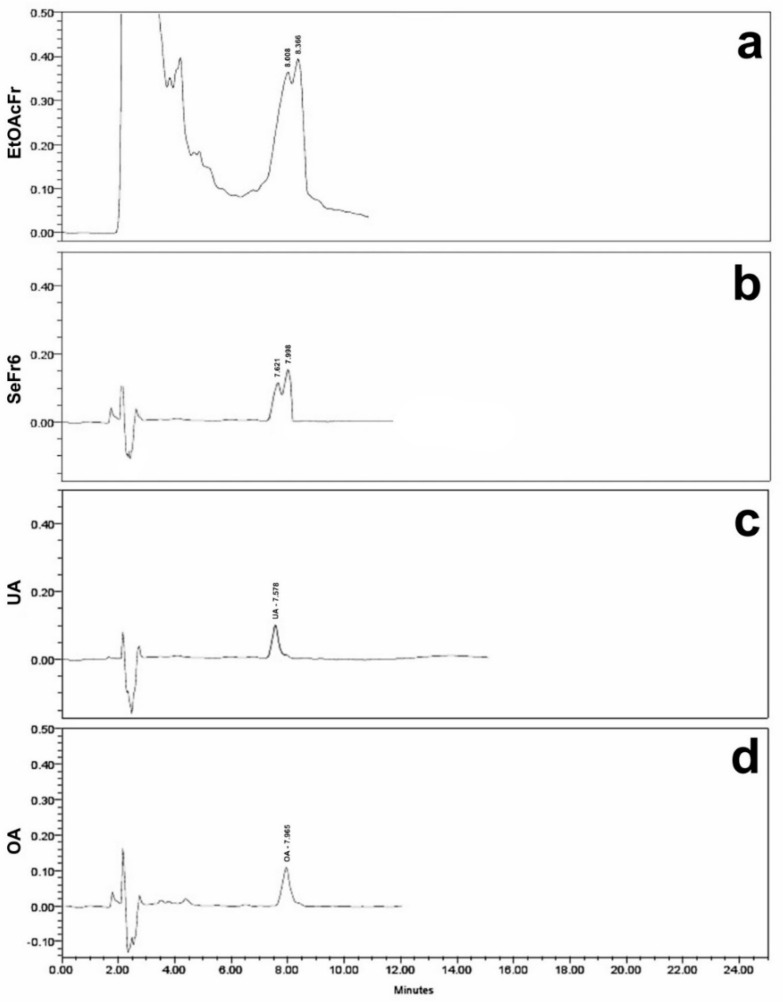
HPLC-DAD chromatograms of EtOAcFr (**a**), SeFr6 (**b**), UA (**c**) and OA (**d**) standard.

**Figure 2 plants-11-00575-f002:**
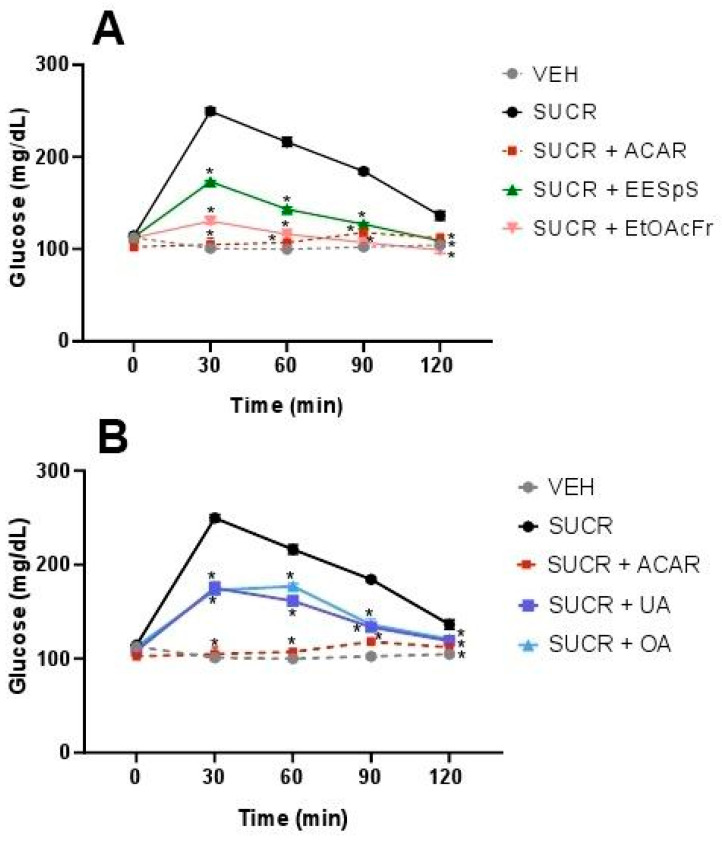
Effect of products obtained from the stems from *S. polystachya* in oral sucrose tolerance test (OSuTT). (**A**) OSuTT of groups treated with sucrose, acarbose, EESpS and EtOAcFr, as well as vehicle. (**B**) OSuTT of groups treated with sucrose, acarbose, UA and OA, as well as vehicle. Results shown as mean ± SEM (*n* = 6). * *p* > 0.05 significantly different vs. sucrose group values at the same time. (Two-way ANOVA followed by Dunnett post hoc test) ACAR: acarbose; SUCR: sucrose; EESpS: ethanolic extract of stem; EtOAcFr: ethyl acetate fraction; UA: ursolic acid and OA: oleanolic acid.

**Figure 3 plants-11-00575-f003:**
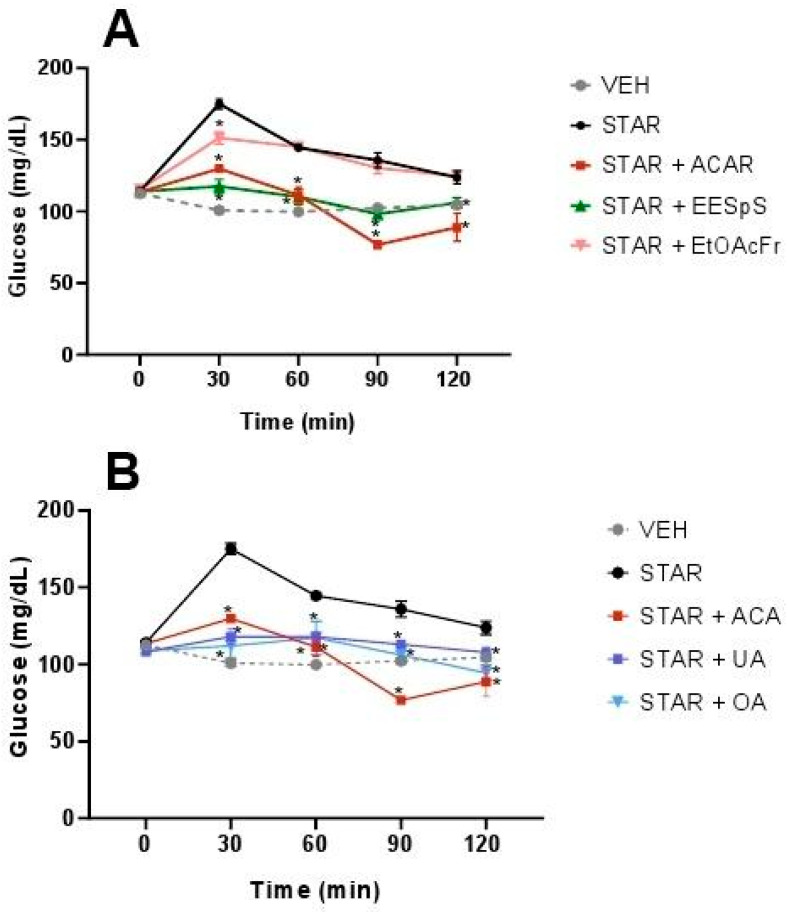
Effect of products obtained from the stems from *S. polystachya* in oral starch tolerance test (OStTT). (**A**) OStTT of groups treated with starch, acarbose, EESpS and EtOAcFr, as well as vehicle. (**B**) OStTT of groups treated with starch, acarbose, UA and OA, as well as vehicle. Results shown as mean ± SEM (*n* = 6). * *p* > 0.05 significantly different vs. starch group values at the same time. (Two-way ANOVA followed by Dunnett post hoc test) ACAR: acarbose; STAR: starch; EESpS: ethanolic extract of stem; EtOAcFr: ethyl acetate fraction; UA: ursolic acid and OA: oleanolic acid.

**Figure 4 plants-11-00575-f004:**
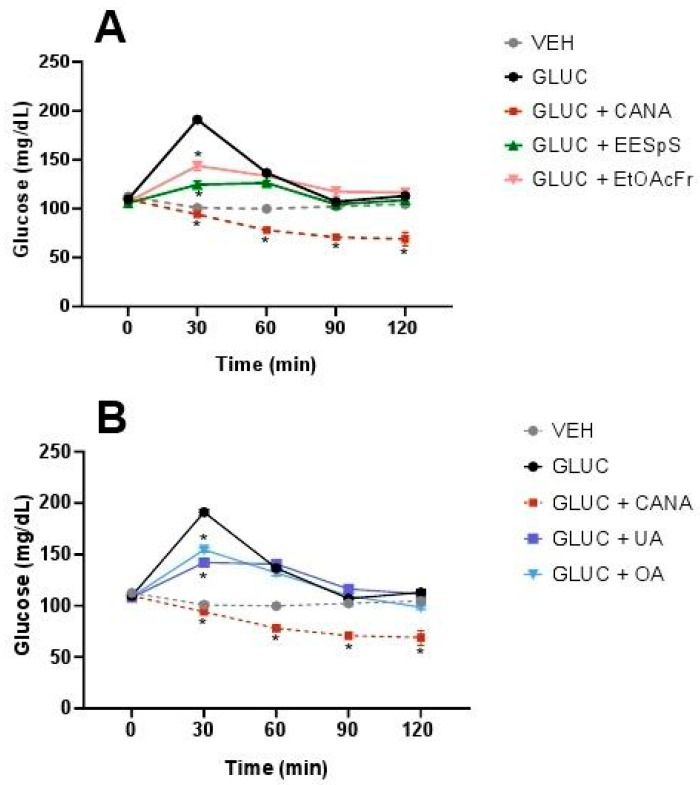
Effect of products obtained from the stems from *S. polystachya* in oral glucose tolerance test (OGTT). (**A**) OGTT of groups treated with glucose, acarbose, EESpS and EtOAcFr, as well as vehicle. (**B**) OGTT of groups treated with glucose, acarbose, UA and OA, as well as vehicle. Results shown as mean ± SEM (*n* = 6). * *p* > 0.05 significantly different vs. glucose group values at the same time. (Two-way ANOVA followed by Dunnett post hoc test) ACAR: acarbose; GLUC: glucose; EESpS: ethanolic extract of stem; EtOAcFr: ethyl acetate fraction; UA: ursolic acid and OA: oleanolic acid.

**Figure 5 plants-11-00575-f005:**
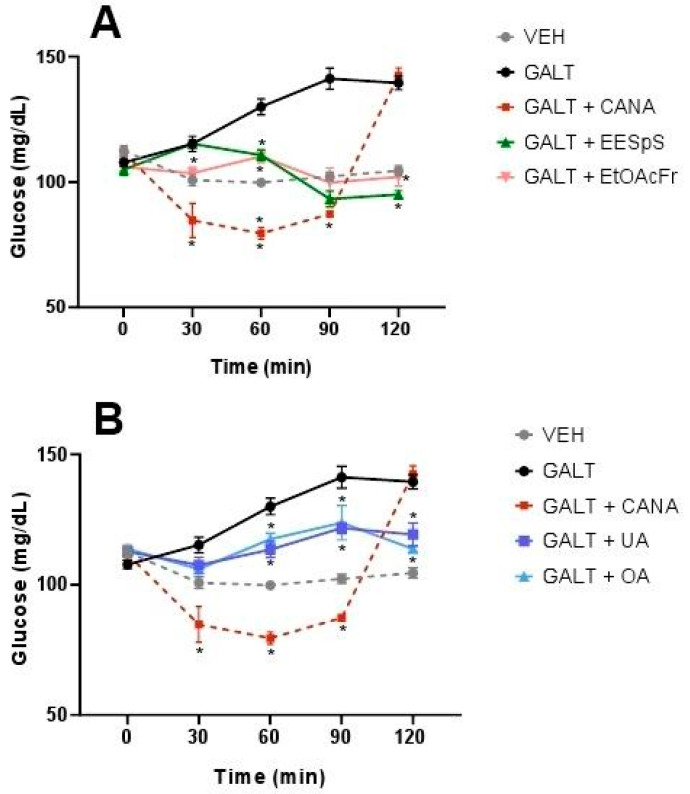
Effect of products obtained from the stems from *S. polystachya* in oral glucose tolerance test (OGaTT). (**A**) OGaTT of groups treated with galactose, acarbose, EESpS and EtOAcFr, as well as vehicle. (**B**) OGaTT of groups treated with galactose, acarbose, UA and OA, as well as vehicle. Results shown as mean ± SEM (*n* = 6). * *p* > 0.05 significantly different vs. galactose group values at the same time. (Two-way ANOVA followed by Dunnett post hoc test) ACAR: acarbose; GALT: galactose; EESpS: ethanolic extract of stem; EtOAcFr: ethyl acetate fraction; UA: ursolic acid and OA: oleanolic acid.

**Figure 6 plants-11-00575-f006:**
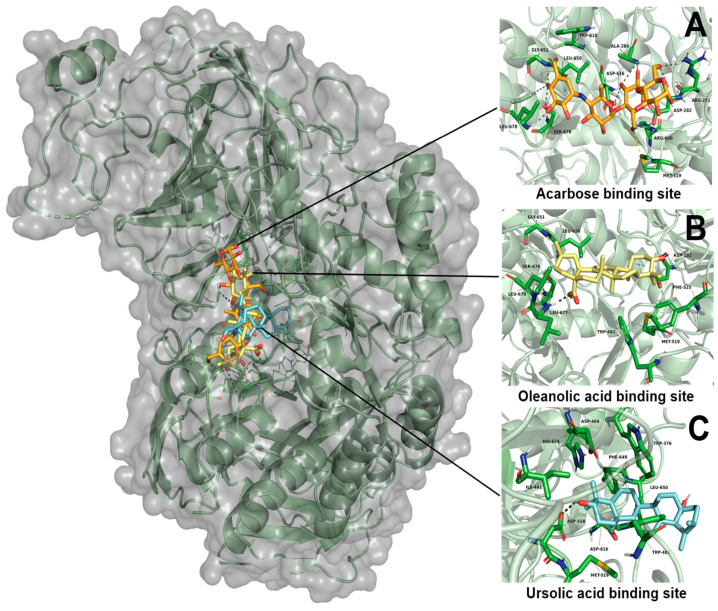
Results of molecular docking on α-glucosidase enzyme. (**A**) Interaction of acarbose and its binding site position; (**B**) interaction of oleanolic acid and its binding site position; (**C**) interaction of ursolic acid and its binding site position.

**Figure 7 plants-11-00575-f007:**
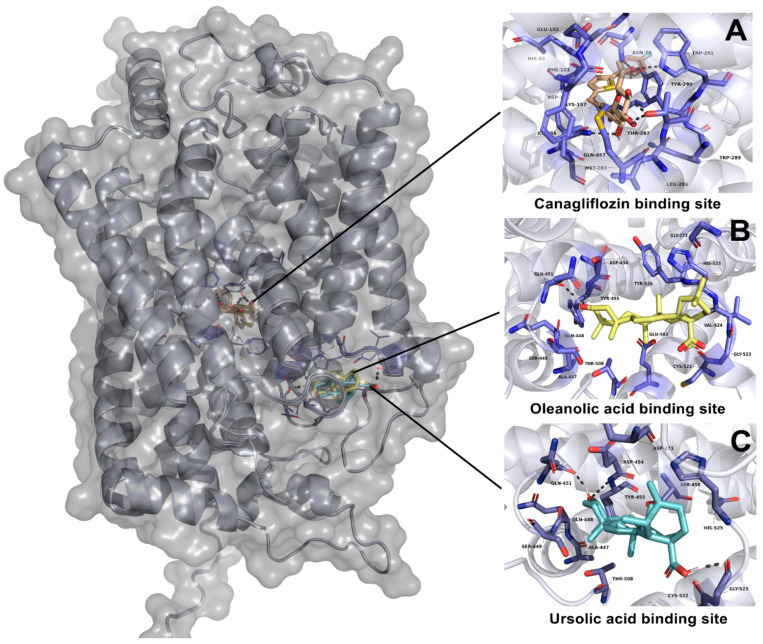
Results of molecular docking on SGLT1 cotransporter. (**A**) Interaction of canagliflozin and its binding site position; (**B**) interaction of oleanolic acid and its binding site position; (**C**) interaction of ursolic acid and its binding site position.

**Table 1 plants-11-00575-t001:** Effect of the acute administration of ethanolic extract of the stems obtained from *S. polystachya* on the body weight and relative organ weight of normal female mice.

Group	Dose (mg/kg)	Pancreas (g)	Liver (g)	Kidneys (g)	Stomach (g)
Normal	-	0.8 ± 0.0	4.9 ± 0.1	1.2 ± 0.05	1.4 ± 0.1
EESpS	50	0.7 ± 0.0	4.5 ± 0.2	1.1 ± 0.1	2.1 ± 0.1
	300	0.6 ± 0.1	5.0 ± 0.3	1.2 ± 0.1	2.0 ± 0.5
	2000	0.8 ± 0.2	4.8 ± 0.2	1.2 ± 0.0	1.9 ± 0.5

Mean ± SEM of relative organ weight (g) of normal female mice (*n* = 3). (ANOVA followed by Dunnett post hoc test) vs. the normal group values. EESpS: ethanolic extract of the stem.

**Table 2 plants-11-00575-t002:** Acute effect of the ethanolic extract of stems from *S. polystachya* and its products over glycemia.

Group	Dose (mg/kg^−1^)	0 min	30 min	60 min	90 min	120 min
NM control	-	175.3 ± 1.4	174 ± 1.8	170.5 ± 2.6	172.8 ± 2	178.5 ± 1.7
DM control	-	302.5 ± 5.7	306 ± 3.7	309.4 ± 2.8	311.5 ± 1.5	294.8 ± 0.6
DM + EESpS	50	295.5 ± 5	310.3 ± 12	290.3 ± 14	300 ± 20	291.8 ± 1
DM + EESpS	100	301.8 ± 1	242.3 ± 0.6 *	258.3 ± 1.2 **	241.3 ± 0.8 ^■^	247.3 ± 0.3 ^■■^
DM + EESpS	200	303.3 ± 10	320.4 ± 7.3	307.8 ± 1.3	287 ± 9.5 ^■^	313.3 ± 3.5
DM + AqRFr	50	309 ± 2.6	268.4 ± 2.7	231.6 ± 3.4	218.8 ± 1	213.3 ± 7.9
DM + EtOAcFr	50	300 ± 4.8	249 ± 5.9 *	203.3 ± 1.8 **	213 ± 4.4 ^■^	187.5 ± 0.6 ^■■^
DM + SeFr6	50	293.5 ± 2.5	338.8 ± 2.2	386 ± 3.9	330.3 ± 4.2	259.3 ± 5.1 ^■■^
DM + Oleanolic acid	50	283.2 ± 5.2	302 ± 11.4	302 ± 7.2	239.4 ± 3.1 ^■^	252.8 ± 2.9 ^■■^
DM + Ursolic acid	50	268.7 ± 4	220.8 ± 5.9 *	239 ± 6.6 **	259.2 ± 4 ^■^	236 ± 5.5 ^■■^
DM + Acarbose	50	308.7 ± 1.6	236.8 ± 2.3 *	237.8 ± 3.4 **	249 ± 1.5 ^■^	225 ± 2.5 ^■■^
DM + Glibenclamide	50	307.6 ± 1.8	243.5 ± 6.2 *	247.8 ± 5.4 **	268 ± 1.9 ^■^	292 ± 2.3 ^■■^

Blood glucose levels, mean ± SEM (*n* = 6). * *p* > 0.05 significantly different vs. DM control 30 min; ** *p* > 0.05 significantly different vs. DM control 60 min; ^■^
*p* > 0.05 significantly different vs. DM control 90 min; ^■■^
*p* > 0.05 significantly different vs. DM control 120 min (ANOVA followed by Dunnett post hoc test). GLIB: glibenclamide; ACAR: acarbose; EESpS: ethanolic extract of the stem from *S. polystachya.*

**Table 3 plants-11-00575-t003:** Quantity of glucose measured in the external aqueous medium, percent inhibition, and CE_50_ calculated after addition of treatments on intestinal sucrose hydrolysis (ISH) inhibition test.

Treatment	Glucose (mg/dL^−1^)	Glucose (mg/dL^−1^)	% of Inhibition	IC_50_
	0 h	2 h		
Sucrose (15%)	0 ± 0	90.6 ± 5.8	-	-
EESpS [200 µgmL^−1^]	0 ± 0	71.6 ± 1.6 *	20.8 ± 1.8	734.3 µgmL^−1^
EESpS [400 µgmL^−1^]	0 ± 0	55 ± 2 *	39.2 ± 2.2	
EESpS [800 µgmL^−1^]	0 ± 0	44 ± 1.8 *	51.4 ± 2	
EtOAcFr [200 µgmL^−1^]	0 ± 0	65.6 ± 1.2 *	27.5 ± 1.3	573.5 µgmL^−1^
EtOAcFr [400 µgmL^−1^]	0 ± 0	51 ± 1.4 *	43.7 ± 1.6	
EtOAcFr [800 µg/mL^−1^]	0 ± 0	35 ± 1.7 *	61.3 ± 1.8	
UA [200 µM]	0 ± 0	59.3 ± 1.8 *	34.5± 2	739.9 µM
UA [400 µM]	0 ± 0	48.3 ± 1.2 *	46.6 ± 1.3	
UA [800 µM]	0 ± 0	45.3 ± 1.9 *	49.9± 2.1	
OA [200 µM]	0 ± 0	87 ± 3.9	3.9 ± 1.6	726.3 µM
OA [400 µM]	0 ± 0	77.3 ± 1.2 *	14.6 ± 1.3	
OA [800 µM]	0 ± 0	50 ± 3.6 *	44.8 ± 4	
Acarbose [200 µM]	0 ± 0	38.2 ± 1.22 *	57.8 ± 1.3	
Acarbose [400 µM]	0 ± 0	13.1 ± 0.75 *	85.5 ± 0.8	173 µM
Acarbose [800 µM]	0 ± 0	7 ± 0.4 *	96.7 ± 0.18	

Effect of products obtained from the stems from *S. polystachya* on intestinal sucrose hydrolysis (ISH) inhibition test. Results shown as mean ± SEM (*n* = 6). * *p* > 0.05 significantly different vs. sucrose group values. (Two-way ANOVA followed by Dunnett post hoc test) ACAR: acarbose; EESpS: ethanolic extract of stem; EtOAcFr: ethyl acetate fraction; UA: ursolic acid and OA: oleanolic acid; IC_50_: half maximal inhibitory concentration.

**Table 4 plants-11-00575-t004:** Quantity of glucose measured in the external aqueous medium, percent inhibition, and CE_50_ calculated after addition of treatments on intestinal glucose absorption (IGA) inhibition test.

Treatment	Glucose (mg/dL^−1^)	Glucose (mg/dL^−1^)	% Of Inhibition	IC_50_
	0 h	1 h		
Glucose (5%)	0 ± 0	217.3 ± 6.1	-	-
EESpS [200 µg/mL^−1^]	0 ± 0	216 ± 8.1	0.5 ± 3.7	1536.3 µg/mL
EESpS [400 µg/mL^−1^]	0 ± 0	192.5 ± 3.7 *	11.4 ± 1.4	
EESpS [800 µg/mL^−1^]	0 ± 0	167.7 ± 9 *	22.8 ± 4.1	
EtOAcFr [200 µg/mL^−1^]	0 ± 0	175.3 ± 19 *	19.3 ± 8.7	697.3 µg/mL
EtOAcFr [400 µg/mL^−1^]	0 ± 0	162.7 ± 1.8 *	25.1 ± 0.8	
EtOAcFr [800 µg/mL^−1^]	0 ± 0	176.5 ± 7.1 *	18.7 ± 7.7	
UA [200 µM]	0 ± 0	270 ± 31.3	0	966.6 µM
UA [400 µM]	0 ± 0	188.3 ± 4 *	13.3 ± 1.8	
UA [800 µM]	0 ± 0	132.3 ± 5.3 *	39.1 ± 2.4	
OA [200 µM]	0 ± 0	189.3 ± 4 *	12.8 ± 1.8	849.3 µM
OA [400 µM]	0 ± 0	132.3 ± 2.2 *	39.1 ± 1	
OA [800 µM]	0 ± 0	121 ± 6.5 *	44.3 ± 3	
Canagliflozin [200 µM]	0 ± 0	230 ± 13.3	0	
Canagliflozin [400 µM]	0 ± 0	113.5 ± 7.4 *	49 ± 3.1	834.5 µM
Canagliflozin [800 µM]	0 ± 0	124 ± 5.7 *	40.5 ± 1.3	

Effect of products obtained from the stems from *S. polystachya* on intestinal glucose absorption (IGA) inhibition test. Results shown as mean ± SEM (*n* = 6). * *p* > 0.05 Significantly different vs. glucose group values. (Two-way ANOVA followed by Dunnett post hoc test) ACAR: acarbose; EESpS: ethanolic extract of stem; EtOAcFr: ethyl acetate fraction; UA: ursolic acid and OA: oleanolic acid; IC_50_: half maximal inhibitory concentration.

**Table 5 plants-11-00575-t005:** Interactions of oleanolic acid (**OA**), ursolic acid (**UA**) acarbose and canagliflozin with residues on the binding sites of α-glucosidase and SGLT1 enzymes.

Compound	α-Glucosidase				SGLT1			
	ΔG (kcal-mol^−1^)	H-BR	NPI	RMSD	ΔG (kcal-mol^−1^)	H-BR	NPI	RMSD
Oleanolic acid	−6.41	Leu677	Asp282, Met519, Phe525, Leu650, Gly651, Ser676, Leu678	-	−10.55	Gln451	Gly272, Ala447, Gln448, Ser449, Asp454, Tyr455, Glu503, Thr508, Cys522, Gly523, Val524, His525, Tyr526	-
Ursolic acid	−5.48	Asp518	Trp376, Trp481, Met519, Asp616, Phe649, Leu650, His674	-	−9.65	Gln451, Gly523	Asp273, Ala447, Gln448, Ser449, Asp454, Tyr455, Ser458, Thr508, Cys522	-
Acarbose	−8.33	Arg281, Asp282, Ala284, Arg600, Asp616, Gly651, Ser676, Leu678	Met519, Trp618, Leu650	1.87	-	-	-	-
Canagliflozin	-	-	-	-	−11.04	Met283, Thr287, Tyr290, Trp291, Gln 457	Asn78, His83, Phe101	1.44

ΔG: Binding energy (kcal/mol^−1^); H-BR: H-binding residues; NPI: nonpolar interactions; Asp: aspartate; Asn: asparagine; Arg: arginine; Gln: glutamine; Lys: lysine; Thr: threonine; Ser: serine; Trp: tryptophan; Leu: leucine; His: histidine; Gly: glycine; Glu: glutamic acid; Ile: isoleucine; Tyr: tyrosine; Phe: phenylalanine.

## Data Availability

The data presented or additional data in this study are available on request from the corresponding author.

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
