# Peer review of "Antihyperglycemic Effects of Salvia polystachya Cav. and Its Terpenoids: α-Glucosidase and SGLT1 Inhibitors"

_plants, 2022, doi:10.3390/plants11050575_

Round 1

Reviewer 1 Report

The article has fundamental and applied significance.The work was done at a sufficient scientific level.There are questions:1-it is unclear why the authors chose glibenclamide rather than canagliflozin as a comparison drug; 2-alpha-glucosidase is a group of enzymes belonging to the class of hydrolases; the paper provides data on the docking of active substrates into alpha-glucosidase, which enzyme was docked specifically?

Author Response

Professor V. Steenkamp Guest Editor (Charles Guo Assistant Editor)      24 January 22                                                                                                                  

of the Special Issue

"Plant Derivatives and Their Pharmaceutical Potential”

Plants 

In agree with editor and referees comments we decided revise the manuscript “Antihyperglycemic Effects of Salvia polystachya Cav. and Its Terpenoids: α-Glucosidase and SGLT1 Inhibitors” by Ortega et al. all changes made are in yellow color, including:

Comments and Suggestions for Authors

Reviewer 1

The article has fundamental and applied significance. The work was done at a sufficient scientific level. There are questions:

Query1

1-it is unclear why the authors chose glibenclamide rather than canagliflozin as a comparison drug;

Answer:

Acarbose an a-glucosidase inhibitor was used as comparative control. In the case of glibenclamide it was used to demonstrated pancreatic b-cells viability in streptozocin-induced diabetic mice in agreement with T2D model

Additional texts were included: Lines 29, 30, 130-132,

Query 2-alpha-glucosidase is a group of enzymes belonging to the class of hydrolases; the paper provides data on the docking of active substrates into alpha-glucosidase, which enzyme was docked specifically?

Answer:

Crystal structure of human lysosomal acid-alpha-glucosidase was used (RCSB PDB-5NN8)

Additional texts were included: Lines358-359, 730-731.

(Crystal structure of human lysosomal acid-alpha-glucosidase was used: RCSB, PDB ID: 5NN8)

Reviewer 2 Report

The manuscript "Antihyperglycemic Effects of Salvia polystachya Cav. And Its 2 Terpenoids: α-Glucosidase and SGLT1 Inhibitors", previously submitted as "Antihyperglycemic Effects of Salvia polystachya Cav. And Its Terpenoids: α-Glucosidase Inhibitors", is interesting, but the changes made are not sufficient. The authors, as suggested by the reviewers, used HPLC analysis, but its performance was not correct.

1. For what purpose the authors used a step gradient up to 25 min. if the analytes had retention times of approx. 8 minutes? What was solvent A and what was B?
2. If the authors used the DAD detector, why did they record the chromatogram only at 220 nm. The registration of the full spectrum allows you to determine the purity of the peak.
3. Performing "room temperature" separation is not recommended. These analytes should be separated at low temperature, eg 10 degrees C. The quality of the obtained separation is the result of several errors made during the analysis. The authors should increase the retention of the analytes to improve their separation (also from other components of the sample). This is very important if we are not using mass detection.

Author Response

Professor V. Steenkamp Guest Editor (Charles Guo Assistant Editor)      24 January 22                                                                                                                  

of the Special Issue

"Plant Derivatives and Their Pharmaceutical Potential”

Plants 

In agree with editor and referees comments we decided revise the manuscript “Antihyperglycemic Effects of Salvia polystachya Cav. and Its Terpenoids: α-Glucosidase and SGLT1 Inhibitors” by Ortega et al. all changes made are in yellow color, including:

Comments and Suggestions for Authors

Reviewer 2

Comments and Suggestions for Authors

The manuscript "Antihyperglycemic Effects of Salvia polystachya Cav. And Its 2 Terpenoids: α-Glucosidase and SGLT1 Inhibitors", previously submitted as "Antihyperglycemic Effects of Salvia polystachya Cav. And Its Terpenoids: α-Glucosidase Query

Inhibitors", is interesting, but the changes made are not sufficient.

The authors, as suggested by the reviewers, used HPLC analysis, but its performance was not correct.

Answer.

This part was corrected two wave lengths were used 220 nm to terpenoids and 254 nm to flavonoids both considering the compounds that previously we isolated of Salvias species (S. amarissima, S. circinata, S. connivens and S polystachya) including linearolactone and polystachynes A, B and D isolated of EtOAcFr of S. polystachya previously. However, any compounds (flavonoids and terpenoids) were detected. When HPLC-DAD analyses were recorded to 254 nm the zone of terpenoids including UA and OA wasn’t present in this sense the maxim UV absorption of the two terpenoids (UA and OA) were 220 nm.

We included a new Figure 1 and additional texts or words were included in lines 149, 150, 162, 163, 457-460, and 602-607.

  1. For what purpose the authors used a step gradient up to 25 min. if the analytes had retention times of approx. 8 minutes? What was solvent A and what was B?

Answer:

Additional text was included lines 602-606.

For elution, a system comprising a binary mobile phase of acetonitrile 100% (A) and acetic acid 2% (B) in water were used. The chromatograph operating conditions were programmed to give the following linear gradient of 80 (A)/20 (B) for 15 min with a flow rate of 0.8 mLmin-1 of mobile phase. The detections were made at a wave length (λ) of 220 nm and 254 nm.

  1. If the authors used the DAD detector, why did they record the chromatogram only at 220 nm. The registration of the full spectrum allows you to determine the purity of the peak.

Answer:

Two wave lengths were used 220 nm to terpenoids and 254 nm to flavonoids both considering the compounds that previously we isolated of Salvias species (S. amarissima, S. circinata, S. connivens and S polystachya) including linearolactone and polystachynes A, B and D isolated previously of EtOAcFr of S. polystachya. However, any compounds (flavonoids and terpenoids) were detected. When HPLC-DAD analyses were recorded to 254 nm the zone of terpenoids including UA and OA wasn’t present in this sense the maxim UV absorption of the two terpenoids (UA and OA) were 220 nm. Also, this zone is free of flavonoids.

  1. Performing "room temperature" separation is not recommended. These analytes should be separated at low temperature, eg 10 degrees C. The quality of the obtained separation is the result of several errors made during the analysis. The authors should increase the retention of the analytes to improve their separation (also from other components of the sample). This is very important if we are not using mass detection.

Answer:

The principal objective of HPLC-DAD was analyzed the SeFr6 obtained of EtOAcFr considering that was the most active. Preliminary TLC analysis showed one purple band, it was compared against terpenoids and flavonoids isolated previously by my group. As results of this TLC the terpenoids UA and OA were consistent. Therefore, both were used as reference. Figure 1 showed that UA and OA are part of SeFr6 in this sense wasn’t necessary low temperature.

May be that other terpenoids typical of Salvia species including polystachynes and linearolactone may are present in SeFr1-5 and SeFr7-10. However, these fractions weren’t actives. Therefore, these fractions were deleted.

Round 2

Reviewer 2 Report

I am not convinced by the Authors' answers. TLC analysis shows nothing of value. The description of the HPLC analysis and its performance are inappropriate. Why use the DAD detector if we do not compare the similarity of the spectra? The detector should register a specific wavelength range, e.g. 190-400 nm. Quantitative analysis is performed at a specific wavelength. Has such an analysis been carried out? Have calibration curves been prepared? Do the concentrations of OA and UA used in biological experiments correspond to the concentration of these compounds in the fractions tested?

Author Response

Estimado revisor, consulte el archivo adjunto.

This manuscript is a resubmission of an earlier submission. The following is a list of the peer review reports and author responses from that submission.

Round 1

Reviewer 1 Report

As a chromatographist, I have some basic comments on the phytochemical part that may affect the interpretation of biological test results:
1) I miss the HPLC-MS phytochemical analysis.
2) If the authors used TLC in this system, they could not find the presence of oleanolic and ursolic acids. These compounds practically do not divide into TLC without prechromatographic derivatization: http://dx.doi.org/10.1016/j.indcrop.2016.09.025

Reviewer 2 Report

The article is of an applied nature.It may be of some interest to a narrow circle of readers.Unfortunately, an unsuccessful design of the study was chosen in the work.The authors claim direct inhibition of alpha-glucosidase by products isolated from a plant source.The methods used give grounds to speak only about indirect action.The study had to be carried out in vitro on the pure enzyme alphaglucosidase, to determine Km and Ki and at the final stage on animals.

Reviewer 3 Report

I reviewed manuscript „Antihyperglycemic Effects of Salvia polystachya Cav. and Its 2 Terpenoids: α-Glucosidase Inhibitors“ submitted to the journal Plants

In a reviewed manuscript authors tested antihyperglycemic effects of Salvia polystachya extract fractions on diabetic and normoglycemic mice. However, authors did not provide information about bioactive compounds in tested extracts so I cannot recommend manuscript for publication. Authors should perform metabolomicsprofiling of the extracts using LC-MS platform. In that way, they will be able to determinate active compounds. Now, they do not know if they have active compounds in their extracts, what is their concentration etc... Without this information results are useless for readers. They use thin layer chromatography (TLC) what is not recommended method for serious publications.